# Antibody–Drug Conjugates in Non-Small Cell Lung Cancer: State of the Art and Future Perspectives

**DOI:** 10.3390/ijms26010221

**Published:** 2024-12-30

**Authors:** Carol Zanchetta, Lorenzo De Marchi, Marianna Macerelli, Giacomo Pelizzari, Jacopo Costa, Giuseppe Aprile, Francesco Cortiula

**Affiliations:** 1Department of Medicine (DAME), University of Udine, 33100 Udine, Italy; zanchetta.carol@spes.uniud.it (C.Z.); demarchi.lorenzo001@spes.uniud.it (L.D.M.); costa.jacopo@spes.uniud.it (J.C.); 2Department of Oncology, University Hospital of Udine, 33100 Udine, Italy; marianna.macerelli@asufc.sanita.fvg.it (M.M.); giacomo.pelizzari@asufc.sanita.fvg.it (G.P.); giuseppe.aprile@asufc.sanita.fvg.it (G.A.); 3Department of Respiratory Medicine, GROW School for Oncology and Reproduction, Maastricht University Medical Centre, 6229 ER Maastricht, The Netherlands

**Keywords:** ADC, NSCLC, target therapy, Trastuzumab deruxtecan

## Abstract

Antibody–drug conjugates (ADCs) represent one of the most promising and rapidly emerging anti-cancer therapies because they combine the cytotoxic effect of the conjugate payload and the high selectivity of the monoclonal antibody, which binds a specific membrane antigen expressed by the tumor cells. In non-small cell lung cancer (NSCLC), ADCs are being investigated targeting human epidermal growth factor receptor 2 (*HER2*), human epidermal growth factor receptor 3 (*HER3*), trophoblast cell surface antigen 2 (*TROP2*), Mesenchymal–epithelial transition factor (*c-MET*), and carcinoembryonic antigen-related cell adhesion molecule 5 (*CEACAM5*). To date, Trastuzumab deruxtecan is the only ADC that has been approved by the FDA for the treatment of patients with NSCLC, but several ongoing studies, both using ADCs as monotherapy and combined with other therapies, are investigating the efficacy of new ADCs. In this review, we describe the structures and mechanism of action of different ADCs; we present the evidence derived from the main clinical trials investigating ADCs’ efficacy, focusing also on related toxicity; and, finally, we discuss future perspectives in terms of toxicity management, possible biomarkers, and the identification of resistance mechanisms.

## 1. Introduction

Lung cancer represents the third most common cause of cancer, and it has the highest mortality rate among all cancers. Non-small cell lung cancer (NSCLC) accounts for about 85% of all lung cancers [1].

Platinum-based chemotherapy has long been the standard treatment for patients with NSCLC. During the last few years, immune checkpoint inhibitors (ICIs) have gained an increasingly important role in lung cancer treatment, improving patients’ survival [2,3,4]. ICIs block the interaction between immune checkpoints and their ligands, activating adaptive immunity and exerting an anti-tumor effect.

From disease progression to first-line treatments, effective therapeutic options are strongly needed, and this is a top research priority.

Antibody–drug conjugates (ADCs) represent a novel class of anti-cancer treatments using a new mechanisms of action. An ADC comprises a linker, an antibody, and a payload. Different combinations of these three elements define a variety of ADCs with different properties and mechanisms of actions [5]. ADCs use an antibody to identify the malignant cells and release the chemotherapy it carries inside of or close to cancer cells [6].

The first approval of an ADC for cancer treatment occurred in 2000 when Gentuzumab ozogamicin, an anti-CD33 mAb-calicheamicin conjugate, was introduced for the treatment of relapsed CD33+ acute myeloid leukemia (AML) [7].

The first approved ADC for solid tumors was Trastuzumab emtansine (T-DM1), an *HER2*-targeted ADC bound to DM1, an agent that prevents microtubule assembly, for the treatment of metastatic breast cancer [8]. In recent years, many ADCs have been investigated for the treatment of solid tumors, and in 2024, another *HER2*-targeted ADC, Trastuzumab deruxtecan (T-DXd), was approved by the FDA for the treatment of *HER2*-positive (IHC3+) metastatic solid tumors after prior systemic therapy if no satisfactory alternative treatment options are available [9].

In this review, we aim to present an overview of the ADCs already approved or under investigation in clinical trials for the treatment of NSCLC. We also describe ADCs’ molecular structure, highlighting the potential advantages of their mechanism of action. Finally, we focus on open questions and challenges to overcome in order to further implement ADCs in clinical practice.

## 2. ADCs’ Structure: The Antibody, the Linker, and the Payload

ADCs consist of three elements: an antibody, a linker, and a payload. The reason ADCs were developed is to conjugate cytotoxic drugs to an antibody that recognizes antigens in the tumor microenvironment in order to release the CT there, thus limiting the side effects, increasing the CT’s efficacy, and increasing the therapeutic window [10]. 

The antibody, binding to the antigen, must have high affinity, potency, and a long half-life (class IGg1 antibodies are generally chosen) [11].

The linker is the component that allows for binding between the antibody and the cytotoxic substance; it should be small enough not to result in steric clutter and allow for detachment of the payload and stable enough not to release the cytotoxic substance into circulation [12]. Linkers consist of several parts: the linker antibody attachment, the chemical trigger, and the linker payload attachment. The linker antibody attachment is the binding site with a specific amino acid residue that allows the linker to be free of steric clutter and flexible in binding to the antibody; the chemical trigger is the portion that allows for the method of cleavage and release of the payload, exploiting the differences in intracellular pH, reduction potential, and enzyme concentration. The linker payload attachment is the element responsible for the final part of the metabolism of the antibody that modifies the kinetics of release of the cytotoxic substance [13].

First-generation ADCs have non-cleavable linkers; they liberate the cytotoxic small molecule payload during lysosomal degradation of the antibody–drug conjugate within the target cell, thus avoiding non-specific release of the drug, while second-generation ADCs have cleavable linkers. In this case, linkers can have different structures: hydrazone, designed for degradation in acid compartments within the cytoplasm, peptide, designed to be enzymatically hydrolyzed by lysosomal proteases, such as cathepsin B, and disulfide, designed to be cleavable through disulfide exchange with intracellular thiols, such as glutathione [14].

The payload identifies the cytotoxic CT that is released into the tumor cell, ideally resulting in tumor cell death. It should be very toxic, stable, and soluble to allow for one of the main mechanisms of action of ADCs, the bystander killing effect, which is made possible by the release of a membrane-permeable payload into the intercellular space, resulting in cytotoxicity on neighboring cells that do not express the specific antigen [15].

Generally, payloads are small molecules that act by inhibiting topoisomerase, such as DXd or SN-38, or the formation of microtubules, such as DM1, leading to cell apoptosis.

New frontiers are represented by payloads, such as biological macromolecules, or radioisotopes, such as iodine-131 and lutetium, which, however, are still being studied in solid tumors (Table 1).

Another important concept is the drug-to-antibody ratio (DAR), which is the average number of drugs linked to each antibody. This is a key element to measure the quality of ADC because it can significantly affect ADC efficacy [16]. In fact, a too low DAR may not deliver enough drug to the target cells, while a too high DAR could lead to increased non-specific toxicity (typical DARs are between two and eight) [17].

The ADCs’ antibody binding to the antigen expressed in tumor cells leads to payload internalization in the endosome and the subsequent release of CT within the tumor cells. Afterwards, the antigen is freed from the antibody, recycled, and re-expressed in the cell membrane, allowing for new internalization [11] (Figure 1 and Figure 2).

ADCs inducing cytotoxic cell death also trigger an immune response, recruiting natural killer cells (NKs) and disrupting the receptor dimerization and, therefore, their replicative function.

This triple action explains the potential of ADCs in the targeted treatment of a growing class of tumors and justifies the reason why they represent the fastest-growing drug classes in oncology [18].

## 3. ADCs in *HER2*-Positive NSCLC

Human epidermal growth factor receptor 2 (*HER2*) is a transmembrane protein encoded by the erb-b2 receptor tyrosine kinase 2 (ERBB2) gene [19]. It belongs to the ErbB or epidermal growth factor receptor (EGFR) family, a group of four cell surface receptors involved in the transmission of signals controlling normal cell growth and differentiation [20]. Activation of downstream signaling pathways occurs through the formation of homodimers or heterodimers. HER2 receptors do not have specific ligands and therefore, most of the time, they function as partners to HER1, HER3, or HER4, to which the ligand is bound [21]. 

In NSCLC, *HER2* represents an actionable genomic alteration (AGA) [22]. Three principal mechanisms of HER2 alterations can be identified: *HER2* gene amplification, *HER2* mutations, and *HER2* protein overexpression [23]. *HER2* mutations are identified in approximately 2% to 4% of NSCLC and are acquired in 1% of *EGFR* TKI-treated patients. *HER2* protein overexpression and gene amplification are present in 6% to 35% and in 10% to 20%, respectively [11]. Immunohistochemistry (IHC) is used for overexpression detection, while next-generation sequencing (NGS) is the gold standard for detecting genetic alterations [24]. 

Anti-HER2 TKIs failed to be effective in NSCLC; in phase II trials, dual EGFR/HER2 TKIs, such as afatinib, and irreversible pan-HER TKIs, like dacomitinib or neratinib, have shown little activity, with ORRs ranging from 0% to 19% [25]. Even selective HER2 TKIs (pyrotinib and poziotinib) have shown disappointing results both in terms of efficacy and safety [26].

### 3.1. Trastuzumab Deruxtecan

Trastuzumab deruxtecan (also known as DS-8201a or T-DXd) is a *HER2*-targeting ADC structurally composed of a humanized anti-HER2 IgG1 monoclonal antibody, a tetrapeptide-based cleavable linker, and a novel cytotoxic topoisomerase I inhibitor (DXd) payload. T-Dxd induces a potent cytotoxic effect (DNA damage and apoptosis of cells) due to its high DAR (approximately eight) and its high membrane-permeability payload, which enables the bystander effect in *HER2*-low tumor cells [6,27].

In the phase II DESTINY-Lung01 trial, they enrolled previously treated patients with unresectable or metastatic NSCLC and *HER2* overexpression (cohort 1, *n* = 90, divided into 1 and 1A according to drug dose of 6.4 mg/kg or 5.4 mg/kg) or *HER2* mutations (cohort 2, *n* = 91).

In the HER2-mutated cohort, the ORR was 55% (1% CR, 54% PR); the median progression-free survival (mPFS) was 8.2 months (95% CI, 6.0–11.9); and median overall survival (mOS) was 17.8 months (95% CI, 13.8–22.1). Among the 33 patients with CNS metastases at baseline, the mPFS was 7.1 months (95% CI, 5.5 to 9.8) and the mOS was 13.8 months (95% CI, 9.8 to 20.9) [28]. 

In the HER2-overexpressing cohort, the ORR was 26.5% in cohort 1 (95% CI 15.0–41.1) and 34.1% in cohort 1A (20.1–50.6). The disease control rate (DCR) was 69.4% in cohort 1 (95% CI 54.6–81.8) and 78.0% in cohort 1A (95% CI 62.4–89.4). In cohort 1 and 1A, the median duration of response (mDoR) was 5.8 months (95% CI 4.3—not evaluable) and 6.2 months (4.2–9.8), respectively; the mPFS was 5.7 months (95% CI 2.8–7.2) and 6.7 months (4.2–8.4), respectively; and the mOS was 12.4 months (95% CI 7.8–17.2) and 11.2 months (8.4—not evaluable), respectively [23,29]. 

Regarding the toxicity profile, more than 90% of patients had one or more treatment-related adverse events (AEs). Approximately 50% of patients experienced toxicity of grade ≥ 3, and the most common grade ≥ 3 AEs were neutropenia, anemia, and fatigue. Of note, pneumonitis of grade ≥ 3 was reported in 8%, 2%, and 18% of patients in cohorts 1, 1A, and 2, respectively [28,29]. 

The Destiny-Lung 02 trial (phase II randomized, non-comparative trial, *N* = 151) evaluated the T-DXd efficacy and safety (patients were randomly assigned to T-DXd 5.4 mg/kg or 6.4 mg/kg). It showed a lower frequency of adjudicated drug-related Interstitial Lung Disease (ILD) (5.9% vs. 14.1%) and a lower frequency of dose interruptions (13.9% vs. 30%) in the 5.4 mg/kg group [30].

According to the results of DESTINY-Lung01 and Lung02 trials, FDA approved T-DXd at 5.4 mg/kg dosing as new standard-of-care treatment for patients with previously treated *HER2*-mutant NSCLC [30].

Current ongoing clinical trials are reported in Table 2. Of note, T-Dxd is also being evaluated as a first-line treatment vs. ICB+CT in patients with advanced non-squamous NSCLC carrying a *HER2* mutation of exon 19 or 20 [31].

### 3.2. Trastuzumab Emtasine

Trastuzumab emtasine is the first *HER2*-targeted ADC developed, and it consists of an anti-HER2 monoclonal antibody (Trastuzumab) linked to a cytotoxic microtubule inhibitor payload DM1 (emtansine) via a non-cleavable thioether linker maleimidyl 4-(N-maleimidomethyl) cyclohexane-1-carboxylate (or SMCC) with a DAR of 3.5 [27,32].

In a phase II trial (*N* = 15), heavily pretreated patients with NSCLC harboring *HER2* exon 20 mutation or overexpression were treated with T-DM1; the ORR, mPFS, and mOS were 6.7% (CI 0.2–32.0), 2.0 months, and 10.9 months, respectively. There was a high percentage of grade 3 or 4 AEs (such as thrombocytopenia and hepatotoxicity) [33].

In a multicenter, prospective, single-arm trial, 49 patients (29 IHC 2+, 20 IHC 3+) received T-DM1 at 6 mg/kg intravenously every 3 weeks. No treatment responses were observed in the IHC 2+ cohort, while the ORR was 20% in the IHC 3+ cohort. In the IHC 2+ and IHC 3+ cohort, the mPFS was 2.6 months (95% CI, 1.4–2.8) and 2.7 months (95% CI, 1.4–8.3), respectively, and the mOS was 12.2 months (95% CI, 3.8–23.3) and 15.3 months (95% CI, 4.1–NE), respectively. Ninety-two percent of patients reported an AE of any grade, 10 patients had grade 3 AEs, and only 1 patient had a grade 4 AE [34]. 

In the subsequent phase II basket trial, 18 patients with advanced NSCLC and HER2 genomic alterations were treated with T-DM1 3.6 mg/kg intravenously every 3 weeks, resulting in an ORR of 44% (95% CI, 22% to 69%) and an mPFS of 5 months (95% CI, 3 to 9 months). Responses were observed in patients harboring *HER2* exon 20 mutation, including two patients who had concurrent *HER2* amplification. Moreover, at this dose, T-DM1 was well-tolerated; no patient stopped therapy as a result of toxicity [35].

## 4. ADCs in TROP2 NSCLC

Trophoblast cell surface antigen 2 (TROP2) is a transmembrane glycoprotein involved in cancer cell growth, proliferation, invasion, and survival [36]. Numerous pieces of evidence show that TROP2 is expressed in various solid tumors, including lung cancer [37].

A systematic meta-analysis (*N* = 16 studies) including 2569 patients with fifteen different solid tumors suggested that TROP2 expression is a poor prognostic factor associated with shorter OS (pooled HR = 1.896, 95% CI = 1.599–2.247, *p* < 0.001) and DFS (pooled HR = 2.336, 95% CI = 1.596–3.419, *p* < 0.001) [38].

### 4.1. Datopotomab Deruxtecan (Dato-DXd)

Datopotomab deruxtecan is a humanized anti-TROP2 IgG with a topoisomerase I inhibitor (deruxtecan) payload linked via a cleavable linker. Although the high stability of the linker in plasma ensures low systemic toxicity, Dato-DXd maintains its cytotoxic power because the ADC is internalized and the payload is released due to the action of the tumor-cell-enriched lysosomal enzymes [27].

A single-arm phase II trial (*N* = 137).) investigated the efficacy of Datopotomab deruxtecan at 6 mg/kg Q3w in patients with advanced or metastatic NSCLC with actionable genomic alterations (*EGFR*, *ALK*, *ROS1*, *NTRK*, *BRAF*, *MET* exon 14 skipping, or *RET*) who have already received at least one line of target therapy and CT. It showed an ORR of 35.8% and a median duration of response (DOR) of 7 months [39].

The phase III RCT TROPION-Lung01 trial compared docetaxel and Dato-DXd in patients with previously treated advanced or metastatic NSCLC after receiving one or two prior lines of therapy. Patients with AGA were eligible. Co-primary endpoints were mPFS and mOS.

At the pre-specified primary analysis, the mPFS was 4.4 months (95% CI, 4.2 to 5.6) and 3.7 months (CI, 2.9–4.2) for Dato-DXd and docetaxel, respectively (HR, 0.75, CI, 0.62–0.91, *p* = 0.004). In patients with non-squamous NSCLC, the HR was 0.63 (CI, 0.51–0.79) compared to 1.41 (CI, 0.92–2.08) in the squamous population. The mOS was 12.9 months (CI, 11.0–13.9) with Dato-DXd and 11.8 months (CI, 10.1–12.8) with docetaxel (HR, 0.94; CI, 0.78–1.14). In patients with squamous histology, the median OS was 14.6 months (95% CI, 12.4 to 16.0) with Dato-DXd and 12.3 months (95% CI, 10.7 to 14.0) with docetaxel (HR, 0.84; CI, 0.68–1.05). However, the OS benefit was not significant even in this population. The most common adverse grade 3 treatment-related adverse events were stomatitis (6.7%), anemia (4.0%), and ILD (3.7%) in the experimental arm and neutropenia (23.4%) in the docetaxel arm.

This trial showed a PFS benefit of Datopotomab deruxtecan over docetaxel but failed to show an OS improvement [40]. Of fundamental importance appears to be patient selection in order to reduce unnecessary toxicities and enhance the benefit seen in the subgroup analysis [41].

The combination of Dato-Dxd and immunotherapy is currently being evaluated; TROPION-Lung07 is testing Datopotomab Deruxtecan with Pembrolizumab with or without platinum chemotherapy in patients with previously untreated advanced or metastatic non-squamous NSCLC with PD-L1 expression < 50% and without actionable genomic alterations [42].

A phase III study (TROPION-Lung08, NCT05215340) is also evaluating the efficacy of first-line Dato-DXd plus pembrolizumab versus pembrolizumab monotherapy in patients with advanced/metastatic NSCLC without actionable genomic alterations and with a PD-L1 tumor proportion score ≥ 50%. Primary endpoints are progression-free survival and overall survival [43].

### 4.2. Sacituzumab Govitecan

Sacituzumab govitecan is a Trop-2-directed antibody–drug conjugate that selectively delivers topoisomerase I inhibitor SN-38 linked by a cleavable linker, an active metabolite of irinotecan [4].

A phase I/II basket trial evaluating the safety of Sacituzumab reported the most occurring TRAEs as nausea (62.6%), diarrhea (56.2%), fatigue (48.3%), alopecia (40.4%), and neutropenia (57.8%). Neutropenia occurred in 57.8% of patients, although febrile neutropenia was infrequent (5.5%). Serious AEs occurred in 38.8% of patients, the most common being febrile neutropenia (4.0%), diarrhea (2.8%), vomiting (1.4%), and nausea (1.2%). Adverse events led to treatment discontinuation in 51.7% of patients [44].

Interestingly, the presence of homozygosity of the UGT1A1 *28 allele (*28/*28; 9.3% of patients) was associated with an increased risk for neutropenia with higher all-grade neutropenia in homozygous patients compared with heterozygous or wild-type patients. Prescreening for UGT1A1 genotype is then not strictly necessary but patients with a homozygous *28/*28 genotype should be closely monitored for an increased risk of neutropenia. 

The phase III EVOKE 01 study (*N* = 603) tested sacituzumab govitecan versus docetaxel in metastatic non-small cell lung cancer after progression on platinum-based CT, anti-PD-L1, or AGA-targeted treatments. Sacituzumab govitecan led to a 16% reduction in risk of death in the ITT population (primary endpoint), but this finding was not statistically significant. The median PFS was 4.1 months (95% CI, 3.0 to 4.4) with SG and 3.9 months (95% CI, 3.1 to 4.2) with docetaxel (HR, 0.92 [95% CI, 0.77 to 1.11]). The median PFS was similar between patients with squamous (3.8 months [95% CI, 2.8 to 5.4]) and non-squamous (4.1 months [95% CI, 2.9 to 5.3]) NSCLC.

The most common grade ≥ 3 TRAEs in the experimental arm were fatigue (12.5%), diarrhea (10.5%), neutropenia (24.7%), and anemia (6.4%). Febrile neutropenia was reported in 7.8% of cases. In the docetaxel arm, 36.8% of patients experienced neutropenia, and 9.4% experienced febrile neutropenia [45].

EVOKE 02 is currently testing SG in combination with pembrolizumab with/without platinum-based chemotherapy in previously untreated metastatic NSCLC lacking actionable genomic alterations [46]. Another trial, EVOKE 03, is comparing SG plus pembrolizumab and pembrolizumab monotherapy in first-line PD-L1 metastatic NSCLC [47].

As seen for Dato-DXd, data from EVOKE trials suggest that precise and meticulous patient selection is required to optimize the benefit of the treatment.

## 5. ADCs in *HER3* NSCLC

*HER3* is a member of the HER family aberrantly expressed in a large number of tumors. *HER3* has impaired kinase activity that requires heterogeneous coupling with various receptors, including *EGFR*, *HER2*, *HER4*, and *MET*, to gain its transphosphorylation and activate downstream signaling pathways [48].

Protein expression of *HER3* has been observed in 83% of patients with NSCLC, and it is associated with metastatic progression and reduced survival [49].

In *EGFR*m NSCLC, HER3 preferentially couples with EGFR, activates the PI3K/AKT signaling pathway, and inhibits apoptosis. Using EGFR-TKIs, HER3 and its downstream pathway become inactive, inducing apoptosis of cancer cells.

In addition, *HER3* plays a crucial role in EGFR-TKI resistance when *MET* genomic amplification is present because through its coupling with *MET*, *HER3* maintains antiapoptotic HER3/PI3K/AKT signaling [48,50].

### Patritumab Deruxtecan

Patritumab deruxtecan is an ADC composed of a humanized anti-HER3 monoclonal antibody (patritumab) conjugated to a topoisomerase I inhibitor payload (DXd, exatecan derivative) via a cleavable tetrapeptide linker, with a DAR of four [51]. 

The phase II HERTHENA-Lung01 trial evaluated HER3-DXd in 225 patients with advanced *EGFR*-mutated NSCLC previously treated with EGFR TKI therapy and platinum-based chemotherapy (PBC).

The ORR was 29.8% (95% CI, 23.9–36.2), the median duration of response was 6.4 months (95% CI, 4.9 to 7.8), the median progression-free survival was 5.5 months (95% CI, 5.1 to 5.9), and the median overall survival was 11.9 months (95% CI, 11.2 to 13.1). In patients with active brain metastases at baseline (*n* = 30), the CNS ORR was 33.3% (95% CI, 17.3–52.8). 

HER3-DXd once every 3 weeks had a manageable safety profile, with a dose interruption in 40.4% of patients (91/225), dose reduction in 21.3% (48/225), and treatment discontinuation in 7.1%. Treatment-emergent adverse events (TEAEs) of grade ≥ 3 and ≥4 severity occurred in 64.9% and 28.9% of patients, respectively, and the most common grade ≥ 3 TEAEs were hematologic toxicities. Twelve patients (5.3%) experienced drug-related ILD (5.3% grade 3–4) [48,52]. 

Based on these results, the phase III HERTHENA-Lung02 trial is investigating the efficacy and safety of HER3-DXd compared with PBC in patients with metastatic or locally advanced NSCLC with a common *EGFR*-activating mutation after progression with a third-generation EGFR TKI [49].

Of note, HER3-Dxd demonstrated clinically meaningful efficacy with durable responses in patients with different *HER3* expression levels, those with different EGFR TKI resistance mechanisms, and those with brain metastases. What would be interesting would be identifying a specific subset of patients who are likely to benefit the most from the HER3-DXd treatment.

## 6. ADCs in MET NSCLC

*c-MET* (Mesenchymal–epithelial transition factor) is a receptor tyrosine-protein kinase encoded by the proto-oncogene MET. The binding of HGF to the MET receptor, and thus its dimerization and its autophosphorylation, favors downstream signaling through the RAS–RAF–MEK–ERK pathway and the PI3K–AKT–mTOR pathway, which promote proliferation, migration, and cellular invasion [53].

In NSCLC, *MET* overexpression (30–50%), *MET exon 14* skipping mutations (3%), and *MET* amplification (1.5%) have been identified as primary driver alterations. The latter has been associated with a resistance mechanism in *EGFR*-mutant NSCLC resistant to EGFR TKIs [21]. 

For patients with *MET exon 14* skipping mutations NSCLC, TKIs, such as capmatinib and tepotinib, received recent FDA approval, while for *MET* overexpression and amplification, no targeted treatment is currently approved [54]. 

### Telisotuzumab Vedotin

Telisotuzumab vedotin (Teliso-V) consists of a humanized monoclonal antibody, telisotuzumab (ABT-700), coupled to the microtubule inhibitor monomethyl auristatin E (MMAE) through a cleavable valine–citrulline linker in a DAR of 3.1 [55]. 

The encouraging results (the ORR was 18.8%, with an mPFS of 5.7) and the tolerated safety profile shown in the phase 1 dose-escalation study (*N* = 16) in solid tumors with *c-MET* overexpression led to the determination of the recommended dose for phase 2 (RP2D) at 2.7 mg/kg every 3 weeks [56]. The phase II SWOG S1400K study evaluated Teliso-V 2.7 mg/kg q3w in patients with advanced treatment-refractory squamous NSCLC with *c-MET* IHC-positivity (defined by an H-score ≥ 150), but the trial was discontinued prematurely due to failure to meet pre-specified response criteria, with an ORR of only 9% (2/23) and severe toxicity (three fatal grade 5 pulmonary TRAEs) [57].

The single-arm phase II LUMINOSITY trial (*N* = 161) enrolled patients with locally advanced or metastatic squamous or non-squamous NSCLC whose tumors were *c-MET* overexpressed by central immunohistochemistry to receive second- or third-line treatment with Teliso-V 1.9 mg/kg every 2 weeks. Results from an interim analysis showed in *c-MET* high expression and *c-MET* intermediate expression patients an ORR of 35% and 23% and an mDOR of 9 months and 7.2 months, respectively. The mOS was similar in both subgroups at 14.6 months and 14.2 months, respectively. 

Concerning toxicity, treatment-related adverse effects were manageable and moderately tolerated; the most common any-grade treatment-related AEs were peripheral sensory neuropathy (30%), peripheral edema (16%), and fatigue (14%); grade 5 TRAEs occurred in two patients [58].

Teliso-V is also being evaluated against docetaxel in the randomized phase 3 TeliMET NSCLC-01 trial in patients with previously treated non-squamous NSCLC who express c-Met and are wild-type for EGFR in combination with osimertinib in the phase 1 M14-237 trial [59]. 

Considering that *MET* amplification represents a common resistance mechanism to EGFR TKIs in *EGFR*-mutant NSCLC, Teliso-V has also been explored in this subgroup of patients. 

The phase Ib study evaluated Teliso-V (2.7 mg/kg once every 21 days) plus erlotinib (150 mg once daily) in *EGFR*-activating mutation and *c-MET*+ NSCLC patients who have progressed to an EGFR TKI. The ORR was 32.1% (with a high ORR in patients with a higher *c-MET* H-score), the DCR was 85.7%, the mPFS was 5.9 months, and the majority of adverse events were low-grade [60].

Currently, patients harboring this AGA have very limited treatment options, and there are currently no approved targeted therapies. Teliso-V, in the LUMINOSITY trial, was associated with durable responses in c-Met protein-overexpressing non-squamous *EGFR*-wildtype NSCLC regardless of the level of expression, giving hope for an approved targeted treatment in the near future [58].

## 7. ADCs in CEACAM5 NSCLC

Carcinoembryonic antigen-related cell adhesion molecule 5 (*CEACAM5*) is a transmembrane glycoprotein that is involved in cell adhesion and migration and is upregulated in different tumor types. High expression of *CEACAM5* is associated with worse survival in patients with non-small cell lung cancer [61].

### Tusamitamab Ravtansine

Tusamitamab ravtansine is an antibody–drug conjugate (ADC) that consists of a humanized monoclonal antibody selective for the extracellular domain of *CEACAM5*, a cleavable disulfide linker, and a microtubule assembly inhibitor as the payload (DM4). The average drug/antibody ratio of the molecule is 3.8 [62].

In a phase I dose-escalation trial (ClinicalTrials ID NCT02187848), patients with an advanced solid tumor received intravenous Tusamitamab ravtansine. It confirmed that the maximum tolerated doses were 170 mg/m^2^ (LD), followed by 100 mg/m^2^ Q2W, and 170 mg/m^2^ Q3W as a fixed dose. Asthenia (21.4% in the Q2W-LD and 26.7% in the Q3W cohort), nausea (21.4% vs. 20.0%), abdominal pain (17.9% vs. 20.0%), and keratopathy (17.9% vs. 20.0%) were the most common treatment-emergent adverse events in both cohorts. Grade 1 peripheral neuropathy was also observed (incidence of 21.4% in the Q2W-LD and 33.3% in the Q3W cohort). Dose discontinuations for AEs occurred in less than 10% of patients in both cohorts. Primary prophylaxis with a vasoconstrictor and a corticosteroid was used to reduce the incidence of corneal events, without a clear benefit [63].

The phase III RCT CARMEN-LC03 evaluated Tusamitamab ravtansine in patients progressing on a second line of treatment. The study compared Tusamitamab ravtansine as monotherapy to docetaxel in patients with metastatic non-squamous NSCLC and high *CEACAM5* expression. An Independent Data Monitoring Committee (IDMC) found that Tusamitamab ravtansine as a monotherapy did not meet its dual primary endpoint of progression-free survival (PFS) compared to docetaxel. An improved overall survival (OS) trend was not considered sufficient for study continuation, and the termination of the program was based on non-improvement in PFS at the final analysis [64].

A phase II study CARMEN-LC04 is evaluating Tusamitamab ravtansine plus ramucirumab in patients that experienced disease progression after an immune checkpoint inhibitor and platinum-based chemotherapy. As of April 2023, 31 total patients were treated, and the mPFS was 5.7 months (CI 95%, 5.4–9.1) with a median exposure to treatment of 24.1 weeks [65].

CARMEN-LC05 is a phase II, open-label, non-randomized study that evaluated Tusamitamab ravtansine in combination with pembrolizumab and pembrolizumab + platinum-based chemotherapy ± pemetrexed in patients with advanced non-squamous NSCLC. As of March 2024, 57 patients were treated with Tusamitamab ravtansine. In the experimental arm, Tusamitamab ravtansine 150 mg/m^2^ + pembrolizumab, the objective response rate (ORR, confirmed complete response and partial response) was 47.8%, and the ORR reached 59.1% in the add-on arm with Tusamitamab ravtansine 150 mg/m^2^ + pembrolizumab + platinum + pemetrexed. The median PFS was 11.6 months (CI 7.7; 15.9), and the median duration of response (DOR) was 12.5 months (CI 8.4; 24.1) in all treated patients. Peripheral neuropathy (35%) and corneal keratitis (28%) were by far the most common TRAEs in the entire population.

With the limit of a small sample, it must be noted that most treatment responses occurred in patients with high *CEACAM5* and/or PD-L1 expression [66].

## 8. Discussion

Technical advances in drug development and a better understanding of the biology of NSCLC have enabled the development of ADCs in NSCLC. In addition to the approval of Trastuzumab deruxtecan for patients with previously treated *HER2*-mutated NSCLC, we have described several ADCs that are in clinical development and their encouraging preliminary data.

However, before they can become a standard of care in the near future, a number of questions remain to be clarified, including toxicity management, the utility of biomarkers, the identification of resistance mechanisms, and the best treatment sequence.

Regarding the toxicity profile, the structure of ADCs should reduce side effects through the release of the payload into target cells. However, AEs were often not negligible, leading, in some cases, not only to treatment discontinuation but also to patient death. These events are probably related to several factors, including premature release of the payload or target expression in normal cells [67].

The most frequent toxicities across different ADCs were payload-specific toxicities, such as anemia, neutropenia, thrombocytopenia, fatigue, nausea, stomatitis, and increased liver transaminases. These toxicities occurred in more than 90% of patients, but more than half were grade 1 or 2 toxicities, and if patients were carefully monitored, they usually did not need to stop treatment. These adverse effects appear to be related to the drug dose. In the DESTINY-Lung02 trial, drug-related AEs were approximately two times as frequent in patients treated with T-Dxd at the highest dose (6.4 mg/kg vs. 5.4 mg/kg) [30].

The most severe and impactful quality of life toxicity that has emerged with the use of ADCs and requires some consideration is ILD. Pulmonary toxicity has been most clearly associated with T-Dxd, but it has also been reported with other ADS, such as *HER3*-DXd and Dato-DXd, suggesting a possible payload-specific toxicity. In a pooled analysis of nine phase 1 and 2 studies with T-DXd, the incidence of ILD/pneumonitis was 15.4%, with a median onset of 5.4 months [68].

The mechanism of T-DXd-related ILD remains largely unknown, but as it is a life-threatening effect, its treatment should be initiated as soon as possible. Generally, ADC should be discontinued, and systemic corticosteroids should be administered at a dose related to the severity of the event. In patients with confirmed grade 1 ILD, the ADC can be restarted only if the event is fully resolved, maintaining the same dose if it resolves in 28 days or with a dose reduction if the resolution takes more than 28 days. Patients with a grade ≥ 2 ILD should be treated with corticosteroids for at least 14 days and then tapered over at least 4 weeks. For T-DXd, given its fatality rate, the current ASCO recommendation suggests discontinuing the drug permanently following grade 2 ILD [69].

For other ADCs, no consensus or standardized guidelines are available to guide clinicians in the diagnostic work-up and optimal treatment of ILD.

Future research should focus on optimizing treatment protocols, especially for grade 2 ILD, to understand whether permanent discontinuation is strictly necessary and assess the risk of a second ILD in patients retreated with the same ADC or another one with a similar lung toxicity profile.

Also, in order to manage toxicities and treatment in the most appropriate way, the influence of previous treatments (e.g., ICB) on the risk of developing ILD, as well as the possibility of developing late-onset toxicities, should be investigated.

Dr. Loriot, in his talk at ESMO 2023, proposed different approaches to mitigating ADC- related toxicities. Some simple clinical interventions, such as response-based dose adjustment or reduced treatment duration, might lead to equal efficacy and less toxicities compared to standard treatment. More studies are needed in this field.

Another more complex but promising attempt to decrease the toxicities is represented by the introduction of bispecific ADC. These molecules have increased specificity for tumor cells, reducing on-target, off-tumor toxicities. One more mechanism that can be explored in the near future is silencing the Fc domain of the antibody, thereby reducing off-target, off-tumor, immune-related toxicities.

Some other technological developments, such as innovative payloads, combined payloads, or improved ADC stability in circulation, require researchers’ attention to make these already promising molecules more efficient and safer [70].

An area of rising clinical importance is certainly the use of biomarkers to select patients and to better predict treatment responses.

As we have seen, ADCs target a tumor-associated antigen, so the higher the expression of the antigen, the more effective the treatment.

However, this has not always been confirmed, especially in mBC, where ADCs have been more under investigation.

In particular, post hoc analyses evaluated SG efficacy according to Trop-2 expression in mBC. Bardia et al. demonstrated that outcomes were numerically higher with SG vs. TPC in patients with high and medium Trop-2 expression, but the number of patients with low Trop-2-expressing tumors was too restricted to weaken the role of the employed Trop-2 cutoffs as predictive of response to SG [71].

Meanwhile, the post hoc analysis by Rugo et al. showed no specific level of Trop-2 expression at which SG showed an improved effect [72].

Also, regarding patients with HR-positive mBC with either HER3-high or -low expression, preliminary results demonstrated no difference in terms of activity. In the phase I trial NCT03260491, patients with EGFR-mutant NSCLC progressing to an anti-EGFR TKI who received HER3-DXd showed responses across a wide range of HER3 membrane expression levels, even if a trend toward increased responses was seen for higher H-scores [73].

These findings, derived from phase I/II trials, need validation in larger randomized clinical trials before drawing a definitive conclusion on the reduced correlation between antigen expression and response to treatment.

These obstacles found in predicting responses to ADCs could be explained not only by the bystander effect but also in the different sensitivities of each tumor histotype to the cytotoxic payload delivered or the drug distribution that can be favored or not by specific histologic characteristics, such as scarce/abundant fibrosis and rich/poor vascularization [74].

The identification of clinically actionable biomarkers is a top research priority to select patients most likely to derive a benefit from the treatment and thus also limiting individual and societal toxicity response.

Integration of circulating tumor DNA (ctDNA) can allow for monitoring longitudinal changes in tumor burden and patients’ mutational profiles, being a useful biomarker to delineate responses and resistance to ADCs [75].

The Herald trial supports the role of ctDNA during ADC treatments. It found that ERBB2 copy number levels detected by ctDNA do not influence T-DXd’s activity in patients with *ERBB2*-amplified advanced solid tumors, whereas the clearance of *HER2* amplification in cfDNA during treatment was associated with a higher ORR (88.0% vs. 22.7%) [76].

Solid data about resistance mechanisms to ADCs are missing. Given their mechanism of action, resistance to ADCs might be due to the alteration of one of these necessary events, including binding to a cell-surface antigen, internalization, catabolism, and transport of the released payload from the endo-lysosomal lumen to the cytoplasm.

Initially, target antigens’ downregulation can hinder the effective binding and/or internalization of the antibody–drug conjugate (ADC) into cells. Once internalized, cells might redirect the lysosomal delivery of the ADC by enhancing the recycling of the ADC-bound antigen complex back to the cell surface or by utilizing different endocytic pathways for ADC transport. If the lysosomal environment, which is crucial for the breakdown of the ADC, is compromised, it can result in reduced processing of the ADC and lower release of the drug from the antibody. Additionally, if the substances released from the ADC need a lysosomal membrane transporter to efficiently enter the cytoplasm, loss-of-function in such a transporter could hinder the accumulation of the drug in the cytoplasm. Changes in drug efflux transporters (such as MDR1 and MRP1), mutations in the drug target (e.g., tubulin mutations), or alterations in pro-survival signaling pathways are also potential characteristics of cells that exhibit resistance to ADCs [77].

In pre-clinical breast cancer models, it has been seen that chronic exposure to a target-directed ADC (TDM1) downregulates target expression (*HER2* expression) with subsequently less ADC binding and internalization in tumor cells. In these resistant cell lines, also noted were upregulation of drug efflux pumps, altered mechanisms of endocytosis with dysregulated ADC trafficking to lysosomes, and reduced proteolytic activity, which obviously leads to decreased efficacy [21,78].

Several strategies are being investigated to overcome potential mechanisms of acquired resistance and to improve the toxicity profile, including the sequential use of ADCs with the same target but different payloads and the combination of ADCs with different payloads (dual-payload ADCs) with different mechanisms of action but synergistic action. Moreover, the association with partner drugs that can modulate the target antigen’s dynamics, such as TKI directed against the same ADC target, or anti-angiogenetics and immunotherapeutic agents are under investigation [23]. 

Concerning the best ADCs treatment sequence, initial data are emerging in patients with breast cancer even though published clinical trials are limited.

A retrospective analysis included patients with *HER2*-negative mBC (*N* = 32) treated with more than one ADC. Each line of ADCs beyond the first was evaluated for the presence of the same “antibody target” or “payload” compared to the prior ADC. 

It has been observed that there was a higher number of progressive diseases in the case of using an ADC with the same antibody target while on treatment with the second ADC (69.2% versus 50.0% when the second ADC targeted a different tumor antigen), and similar differences were noted based on the payload switch [79].

Similar data emerged in a recent publication by Mai et al., in which patients with metastatic breast cancer treated with Sacituzumab govitecan and Trastuzumab deruxtecan were included. The clinical activity of both Sacituzumab govitecan and T-DXd appeared modest in patients who had already received an ADC, probably due to cross-resistance to the payload [80].

## 9. Conclusions

In NSCLC, ADCs show promising results and have a strong preclinical rationale. However, more data derived from RCTs are needed before further implementation of their use in clinical practice. Preclinical research and the development of biomarker-oriented RCTs will be key for optimizing ADCs’ development.

## Figures and Tables

**Figure 1 ijms-26-00221-f001:**
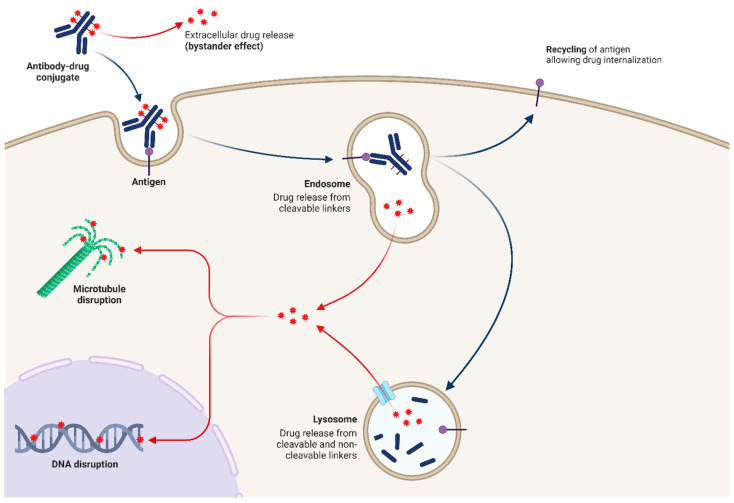
Mechanism of action of ADCs.

**Figure 2 ijms-26-00221-f002:**
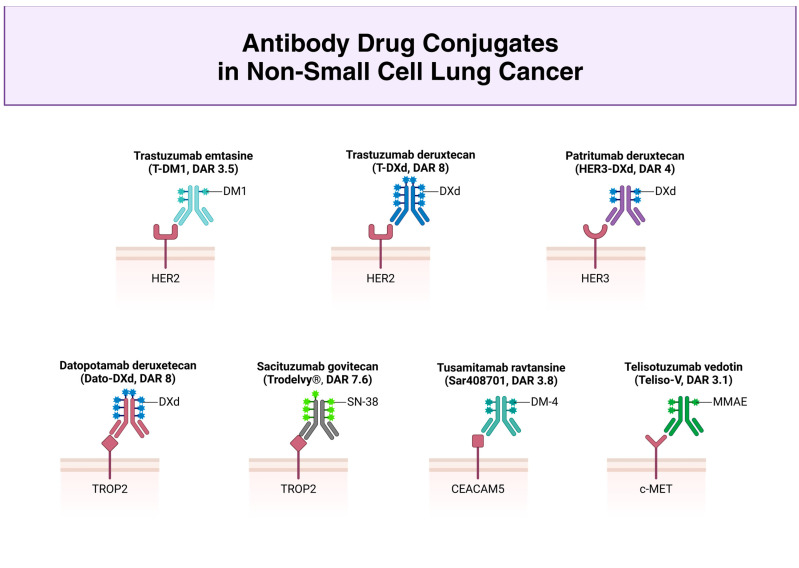
Main ADCs in NSCLC: focus on their structure and DAR.

**Table 1 ijms-26-00221-t001:** Summary of the structure of the main antibody drug conjugates (ADCs) in Non-Small Cell Lung Cancer (NSCLC) patients.

ADC	Antibody Target	Linker	Payload	Payload Action
Trastuzumab deruxtecan	HER2	Cleavable	DXd	Topoisomerase I inhibitor
Trastuzumab emtasine	HER2	Non-cleavable	DM1	Microtubule inhibitor payload
Datopotomab Deruxtecan	TROP2	Cleavable	DXd	Topoisomerase I inhibitor
Sacituzumab govitecan	TROP2	Cleavable	SN-38 (active metabolite of irinotecan)	Topoisomerase I inhibitor
Patritumumab deruxtecan	HER3	Cleavable	DXd	Topoisomerase I inhibitor
Telisotuzumab vedotin	MET	Cleavable	Monomethyl auristatin E (MMAE)	Microtubule inhibitor
Tusamitamab ravtansine	CEACAM5	Cleavable	DM4	Microtubule inhibitor

**Table 2 ijms-26-00221-t002:** Ongoing clinical trials evaluating the efficacy and safety of antibody drug conjugates (ADCs) targeting anti HER2, TROP2, HER3, c-MET, and CEACAM5 in metastatic Non-Small Cell Lung Cancer (NSCLC) patients.

ADC	Target	Study	Fase	Population
Trastuzumab deruxtecan	HER2	Destiny-Lung 02 (Trastuzumab deruxtecan 5.4 or 6.4 mg/kg/21 days)	II	HER2-mutated metastatic NSCLC with recurrence or progression during/after at least one regimen of prior anti-cancer therapy that must have contained a platinum-based Ct drug
Datopotecan Deruxtecan	TROP2	TROPION-Lung07 (Dato-DXd + pembrolizumab + platinum Ct vs. pembrolizumab + platinum Ct and pemetrexed)	III	Advanced/metastatic NSCLC with a PDL1 TPS < 50%
		TROPION-Lung08 (Dato-DXd plus pembrolizumab vs. pembrolizumab)	III	Advanced/metastatic NSCLC with a PDL1 TPS ≥ 50%
Sacituzumab govitecan	TROP2	EVOKE 01 (Sacituzumab govitecan vs. Docetaxel)	III	After progression under platinum Ct and checkpoint inhibitor therapy
		EVOKE 02 (Sacituzumab govitecan plus pembrolizumab ± platinum Ct)	II	Naïve metastatic NSCLC, TPS < 50% or ≥50%
Patritumab deruxtecan	HER3	HERTHENA-Lung01 (Patritumab deruxtecan 5.6 mg/kg fixed dose or an up-titration dose regimen)	II	Previously treated metastatic/locally advanced EGFR-mutated who have progressed on or after at least 1 EGFR TKI and 1 platinum-based Ct
		HERTHENA-Lung02 (Patritumab deruxtecan 5.6 mg/kg/21 days vs. platinum-based Ct)	III	Metastatic NSCLC with EGFR alteration following one or two EGFR TKIs
Telisotuzumab vedotin	c-MET	TeliMET NSCLC-01 (Telisotuzumab vedotin vs. Docetaxel)	III	Patients with metastatic NSQ NSCLC EGFR wild-type and c-MET+ who progressed on at least one systemic therapy
Tusamitamab ravtansine	CAECAM5	CARMEN-LC03 (Tusamitamab ravtansine (100 mg/m^2^/14 days) vs. Docetaxel)	III	Metastatic NSQ NSCLC CEACAM5 with high expression following checkpoint inhibitor plus platinum-based Ct

HER2: Human epidermal growth factor receptor 2; HER3: Human epidermal growth factor receptor 3; TROP2: Trophoblast cell surface antigen 2; C-MET: Mesenchymal–epithelial transition factor; CEACAM5: Carcinoembryonic antigen-related cell adhesion molecule 5.

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
