# Peer review of "Antibody–Drug Conjugates in Non-Small Cell Lung Cancer: State of the Art and Future Perspectives"

_ijms, 2024, doi:10.3390/ijms26010221_

Round 1
Reviewer 1 Report
Comments and Suggestions for Authors
In the manuscript, the author summarized the recent development of Antibody-drug conjugates in non-small cell lung cancer and pointed out the potential future direction. Therefore, I think this article has reached the level of review paper required by International Journal of Molecular Sciences. However, there are still some issues that should be further addressed before publication.
1: Please make the general list for Abbreviation
2: Please exhibit the structure of payload, which are small molecules, and the linker structures. The author should discuss the influence of the linker on the ADC activities, especially the details for structure-activity relationship.
Comments on the Quality of English Languagegood
Author Response
We would like to re-submit the review entitled Antibody-drug conjugates in non-small cell lung cancer: state of the art and future perspectives, after we have addressed your valuable comments.
All the changes we have made are in red. Thank you for recognizing our effort in summarising what are the recent developments on ADCs and what are the possible topics of study to better exploit the action of ADCs.
We are grateful for all your comments that we believe truly contributed in implementing our manuscript.
Here below our point to point answers to your comments.
- Please make the general list for Abbreviation
Thank you for your comment. We implemented our review with the general list for abbreviation as requested, at the end of the text (Page 11-12).
- Please exhibit the structure of payload, which are small molecules, and the linker structures. The author should discuss the influence of the linker on the ADC activities, especially the details for structure-activity relationship.
Thank you very much for your comment. We implemented our section ADCs’ structure: the antibody, the linker and the payload accordingly. We have explained in more detail the three components of ADCs, in particular the linker, describing the possible conformations, its action, the types of cleavage that allow the release of the payload, and the payload, which are small molecules that act by inhibiting topoisomerase or the formation of microtubules leading to cell apopstosis, but we have also introduced new frontiers such as biological macromolecules or radioisotopes such as iodine-131 and lutetium, which, however, are still being studied in solid tumours. We have also better described ADCs' mechanism of action, which consists of several steps, also illustrated in Figure 1, as well as possible actions on the immune system and on the replicative function of receptors.
Page 2, lines 67-99 and 109-111.
We have also reformated the reference list with according to the editor requests.
We also check the manuscript with a native speaker in order to improve the manuscript fluency
Reviewer 2 Report
Comments and Suggestions for Authors
The paper "Antibody-Drug Conjugates in Non-Small Cell Lung Cancer: State of the Art and Future Perspectives" by Carol Zanchetta et al. is intriguing. ADC's function in cancer treatment is a topic of much discussion. To increase its value, more integration and literature review will be necessary.
1. The second section of the manuscript should be expanded to provide more information about antibody selection and the ADC mechanism for cancer treatment, specifically lines 86–90, based on your abstract.
2. As toxicities cross ADCs, you listed payload-specific toxicities, ILD, etc any suggestions for future Research?
3. You mentioned that high expression of cancer antigens is not always a sign of more successful treatment outcomes. What are the potential causes for this? Any recommendations for further research?
4. The topic of acquired resistance is frequently discussed. Could you elaborate further?
Author Response
We would like to re-submit the review entitled Antibody-drug conjugates in non-small cell lung cancer: state of the art and future perspectives, after have addressed your valuable comments.
We are deeply thankful for the time you put in this peer review and we believe your comment truly helped us in improving the manuscript.
All the major changes we have made are in red.
Here below our point to point answers to your comments.
- The second section of the manuscript should be expanded to provide more information about antibody selection and the ADC mechanism for cancer treatment, specifically lines 86–90, based on your abstract
Thank you very much for your comment. We implemented our section accordingly. We have explained in more detail the three components of ADCs, in particular the linker, describing the possible conformations, its action, the types of cleavage that allow the release of the payload, and the payload, which are small molecules that act by inhibiting topoisomerase or the formation of microtubules leading to cell apopstosis, but we have also introduced new frontiers such as biological macromolecules or radioisotopes such as iodine-131 and lutetium, which, however, are still being studied in solid tumours. We have also better described ADCs' mechanism of action, which consists of several steps, also illustrated in Figure 1, as well as possible actions on the immune system and on the replicative function of receptors.
Page 2, lines 67-99 and 109-111.
- As toxicities cross ADCs, you listed payload-specific toxicities, ILD, etc any suggestions for future Research?
Thank you for your comment. We implemented the discussion with a dedicate section accordingly, trying to explore possible strategies to reduce toxicities.
Page 9, lines 470-481.
- You mentioned that high expression of cancer antigens is not always a sign of more successful treatment outcomes. What are the potential causes for this? Any recommendations for further research?
Thank you for youcr comment. We agree with you that this point was not clear. We tried to clarify pur point better and implemented the section triyng to explain why the ADC efficacy is not always proportional to antigen expression.
Page 9-10, lines 485-511.
- The topic of acquired resistance is frequently discussed. Could you elaborate further?
Thank you for the comment. We implmented the manuscript accordingly trying to explore the steps if ADC mechanism of action and highlighting how resistance mechanism can emerge during this process generating drug resistance.
Page 10, lines 524-540
We have also reformated the reference list with according to the editor requests.
We also check the manuscript with a native speaker in order to improve the manuscript fluency. These changes are not in red in order to ease the readibility, but we have a tracked version if you need it.
Round 2
Reviewer 1 Report
Comments and Suggestions for Authors
Accept in present form